Review

# Divergent destinies: insights into the molecular mechanisms underlying EPI and PE fate determination

Paraskevi Athanasouli🄳, Tijs Vanhessche🄳, Frederic Lluis🄳

**Mammalian pre-implantation development is entirely devoted to the specification of extra-embryonic lineages, which are fundamental for embryo morphogenesis and support. The second fate decision is taken just before implantation, as defined by the epiblast (EPI) and the primitive endoderm (PE) specification. Later, EPI forms the embryo proper and PE contributes to the formation of the yolk sac. The formation of EPI and PE as molecularly and morphologically distinct lineages is the final step of a multistage process, which begins when bipotent progenitor cells diverge into separate fates. Despite advances in uncovering the molecular mechanisms underlying the differential transcriptional patterns that dictate how apparently identical cells make fate decisions and how lineage integrity is maintained, a detailed overview of these mechanisms is still lacking. In this review, we dissect the EPI and PE formation process into four stages (initiation, specification, segregation, and maintenance) and we provide a comprehensive understanding of the molecular mechanisms involved in lineage establishment in the mouse. In addition, we discuss the conservation of key processes in humans, based on the most recent findings.**

## Introduction

Mammalian pre-implantation development is entirely devoted to the specification of extra-embryonic lineages and their segregation from embryonic lineage ascendants. Signaling pathways and transcription factors (TFs) form interconnected networks that orchestrate the fate of every cell. Despite the advances in our understanding of how cells make cell fate decisions in mouse pre-implantation development, many fundamental questions remain to be answered. Important aspects in the field include how the emergence of lineage-specific transcriptional programs occurs at early stages, how specific factors influence the differential patterns of expression and consequently the cell fate decision-making process, and which are the signaling pathways acting

upstream of the transcriptional networks that lead to lineage-specific gene activation. Understanding the molecular links between signal transduction, physical cues, and gene expression that dictate cell fate allocation is crucial in elucidating the molecular mechanisms involved in lineage establishment. In this review, we provide a comprehensive overview of the molecular mechanisms, including the transcriptional networks, the epigenetic modifiers, and the signaling pathways that have been shown to be involved in the second lineage decision during mouse pre-implantation development.

## Mouse Embryo: Pre-Implantation Tales

Fertilization marks the onset of pre-implantation development, which extends until the implantation of the embryo in the maternal uterus. After fertilization, the mouse zygote undergoes the first embryonic cleavage (16–20 h after fertilization), giving rise to the two-cell-stage embryo at embryonic day (E) 1.5 (Fig 1). Initially, zygotic development is governed by maternal mRNA, but by the two-cell stage, transcription from the zygotic genome, known as zygotic genome activation, peaks.

After a few rounds of cell division, the 8-cell stage embryo undergoes compaction, where intercellular adhesion is increased, and clear boundaries between neighboring cells are not visible (Fig 1). Concurrently, cells begin to polarize along their apicobasal axis, a process completed by the 16-cell stage. Afterwards, cells start dividing asymmetrically for the next two division rounds, and this results in the generation of two distinct cell types: a polarized outer cell layer, namely, the trophectoderm (TE) lineage, and an apolar inner group of cells, the so-called inner cell mass (ICM). Both cellular populations become fully committed and segregated to the TE and ICM at around the 32-cell stage, which marks the first cell lineage specification (Fig 1). This coincides with the process of cavitation, where a fluid-formed cavity, the blastocoel, forms, and the ICM is driven to the one side of the embryo surrounded by the TE (1). At this stage (E3.25), the embryo is called the blastocyst, and the two cell lineages are characterized by unique TF expression; TE is marked by CDX2, and the ICM uniformly expresses SOX2 (2). OCT4 and NANOG are restricted to the ICM only at the late blastocyst

Department of Development and Regeneration, Stem Cell Institute, KU Leuven, Leuven, Belgium

Correspondence: paraskevi.athanasouli@kuleuven.be; frederic.lluisvinas@kuleuven.be

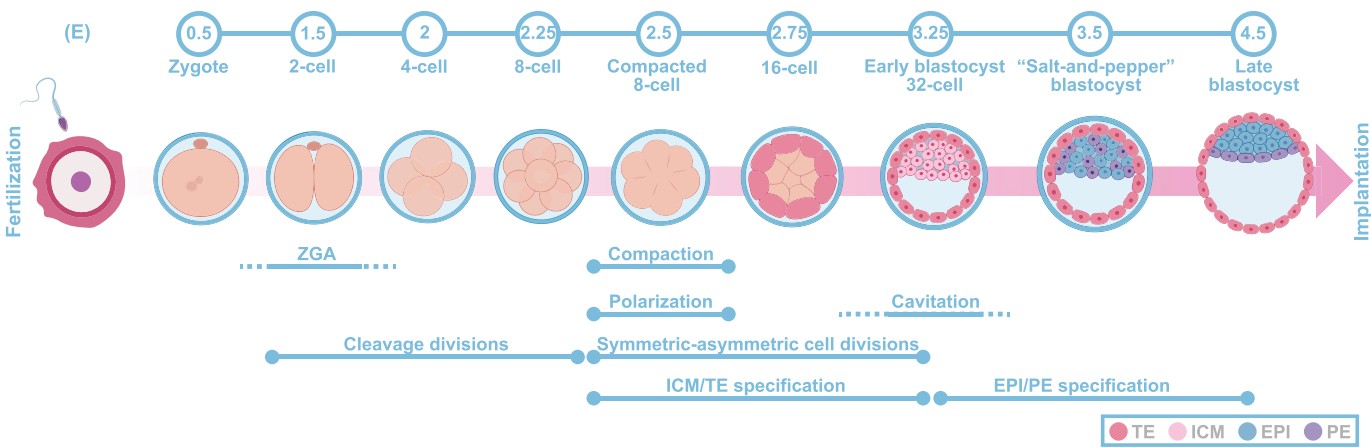

**Figure 1.  Timing and major events in mouse pre-implantation development.**
Upon fertilization, the zygote undergoes the first cleavage, which co-occurs with the zygotic genome activation, followed by two more rounds of cleavage to become an 8-cell embryo. At E2.5, compaction occurs, and the cells become polarized. At that moment, the cells start dividing symmetrically and asymmetrically, leading to TE and inner cell mass formation. After cavity formation, at E3.5, inner cell mass cells are distributed in EPI and PE precursors, which at E4.5 are sorted and allocated into their ultimate localization. Icons were obtained from BioRender.com.

stage (3, 4). The main signaling pathway driving the TE/ICM segregation in mouse embryos is the Hippo/YAP signaling cascade, which has been particularly implicated in TE-specific gene expression (5, 6, 7). TE is the first specified extra-embryonic cell lineage, which plays the primary role in embryo implantation by invading the uterine tissue and later contributes to the placenta (8). In contrast to mice, in humans, distinct transcriptional programs for TE/ICM are only detectable in E5 blastocysts with a discernible blastocoel (9, 10, 11, 12). Moreover, human E5 TE cells are not yet committed and can alter their developmental trajectory when manually positioned internally (13). This flexibility may be associated with the prolonged expression of pluripotency-associated genes such as POU5F1, SOX2, and SALL4 in human TE, unlike mice (9, 10, 14, 15).

After the first lineage specification, murine ICM cells commit to two populations, the epiblast (EPI) or the primitive endoderm (PE) cells, a process known as the second lineage specification. This pivotal process, beginning at the 32-cell stage (E3.25), involves the binary specification of bipotent ICM cells into EPI or PE, the establishment of lineage-specific gene regulatory networks, and the physical sorting of these cells into two distinct layers (Fig 1).

PE forms an epithelium, which is in direct contact with the blastocoel and encloses the EPI cluster between itself and the TE. At this stage (E4.5), EPI and PE hold unique gene expression profiles, with EPI cells expressing primarily *Nanog* and PE cells dominantly marked by *Gata6*'s expression (16, 17, 18). PE is the second extra-embryonic layer specified. After implantation, PE cell derivatives comprise the visceral and the parietal yolk sacs and contribute to gut endoderm formation. As development proceeds, the EPI will form most of the embryo proper, along with the extra-embryonic mesoderm and the amnion. Although the sequential specification of the three (TE, EPI, and PE) blastocyst lineages is well documented in murine development, it was initially hypothesized that the distinct transcriptional states associated with TE, EPI, and PE in humans concurrently emerge by E5 (19). Recent studies have

refined this model, supporting the traditional bifurcation events after identifying a population that might represent an intermediate state between EPI and PE. This population exhibits heterogeneous expression of both lineage markers, suggesting an early ICM (20). Specification of the human ICM into EPI or PE is now believed to occur between the early and mid-stages of the D5 blastocyst, accompanied by the establishment of distinct transcriptional states similar to those observed in mice (21 *Preprint*).

## EPI and PE in a Dish: In Vitro Models

Modeling mouse pre-implantation development has been challenging for developmental biologists, but it is also an imperative need to robustly study early development. The establishment of in vitro cultured stem cell models of all three lineages (TE, EPI, and PE), which constitute the pre-implantation blastocyst, was a breakthrough in our understanding of mouse embryogenesis.

Mouse embryonic stem cells (mESCs) are derived from the ICM of the E3.5 pre-implantation blastocyst and display unlimited proliferation capacity in vitro (22, 23). mESCs are defined as pluripotent because they maintain lineage potential to differentiate into all embryonic germ layers in both in vitro and in vivo assays (16, 18, 24). They are routinely cultured in the presence of serum and the cytokine leukemia inhibitory factor (LIF), which conditions we refer to as naive (25, 26). Transcriptional and functional analyses of mESCs, such as the ability to colonize embryonic, but not extra-embryonic, lineages (16, 18, 24, 25), demonstrate their resemblance to pre-implantation EPI. However, naive mESC cultures in serum+LIF conditions are heterogeneous and mirror the E4.5 EPI cellular composition because they consist of dynamic subpopulations (27, 28, 29, 30, 31, 32). Tracking the expression of PE-specific genes has enabled the identification of PE-biased populations in self-renewing conditions (28, 30, 31).

Extra-embryonic endoderm (XEN) cells are derived from the PE; they are self-renewing, and they are restricted to PE-derived tissues, when injected back into the embryo (33). XEN cells in culture express high levels of PE genes such as *Pdgfra*, *Gata6*, *Gata4*, and *Sox17*; however, they also express markers of the two major derivatives of PE, the visceral (VE) and parietal (ParE) endoderm. Precisely, XEN cells transcriptionally resemble the ParE more and predominately contribute to this layer (33, 34, 35). Yet, the culture of these cells in the presence of BMP4 drives them to a VE fate, indicating that the different stages of the extra-embryonic endoderm (ExEn) can be successfully captured in vitro. The acquisition of pre-implantation–like XEN cells was achieved with the supplementation of medium with activin A and WNT3a, which resulted in the generation of naive, extra-embryonic endodermal progenitor (nEnd) cells (36). The nEnd cell expression profile correlates with E3.5 and E4.5 PE, unlike XEN cells, which correlate with PE only at E4.5, and nEnd cells can contribute both to ParE and to VE, indicating that chemically defined conditions can stabilize their lineage-derived functional characteristics. More recently, primitive endoderm stem cells (PrESCs) were efficiently derived from mouse blastocysts when the latter were cultured in the presence of GSK3-inhibiting conditions along with the FGF4 and PDGF-AA ligands, known to promote PE expansion. PrESCs capture the functionality of E4.5 PE better than XEN cells, because they efficiently incorporate in the PE layer once injected into blastocysts, as well as equally contribute to the PE-derived tissues, VE and ParE (37).

mESCs have provided an excellent model over the years for studying EPI versus PE specification because of their intrinsic heterogeneity and because they can spontaneously transit toward a PE-like state when the expression of key regulators of either program is altered. Disruption of EPI-specific TFs such as *Nanog* or *Prdm14* expression in mESCs results in PE gene activation, therefore specifying cells toward a PE fate (38, 39). On the other hand, the forced expression of *Gata6* or *Gata4* causes the downregulation of *Nanog* and other pluripotency markers, and it is sufficient to induce PE specification in vitro (40, 41, 42). The overexpression of *Sox17* was also capable of driving ExEn fate, with cells mostly resembling the ParE layer (43). These findings indicate that the EPI and PE programs antagonize each other, suggesting that either activation or repression of any of the respective lineage–transcriptional programs may provide the basis for the binary EPI or PE cell fate specification.

## EPI and PE: Molecular Blueprints

GATA6 is widely recognized as a pivotal factor in initiating PE specification by activating the PE transcriptional network (44, 45). In mice, the exclusive expression of *Gata6* in PE progenitors is followed by the sequential activation of the TFs *Sox17*, *Gata4*, and *Sox7*, which collectively facilitate the maturation of the PE lineage (46). However, the role of GATA6 in human development differs significantly from that in mice. In human pre-implantation development, GATA6 is a generic marker for extra-embryonic tissues rather than being PE-specific (47, 48, 49, 50). Moreover, unlike mice, SOX7 is completely absent in human PE. Interestingly, OTX2, typically a

post-implantation EPI and VE marker in mice, is restricted in the human PE during the blastocyst stage (9, 47, 51). This expression shows a temporal and spatial pattern similar to that of murine *Gata6*, indicating significant species-specific differences in gene regulation during early development. Recent studies have begun to clarify the pattern of sequential activation of PE-associated factors in human blastocysts, with PDGFRA emerging as the earliest expressed factor (48).

Unlike the presumptive PE, where *Gata6* is primarily expressed, *Nanog* is not the first EPI factor expressed in the mouse embryo. *Oct4* (*Pou5f1*) is the earliest pluripotency-related TF expressed, beginning at the two-cell stage and continuing throughout cleavage. *Oct4* expression is progressively restricted and, by the expanded blastocyst stage, is confined to the ICM (52). However, *Oct4* is dispensable for *Nanog* expression because *Oct4* null embryos robustly express *Nanog* in the presumptive ICM cells (53). Similarly, *Sox2* is weakly expressed at the two-cell stage, and its expression becomes augmented and nuclear at the 8-cell stage, whereas at the blastocyst stage, all ICM cells express *Sox2* (54). Maternal and zygotic deletion of *Sox2* in mouse embryos did not affect the percentage of NANOG+ cells until E3.75, indicating that *Sox2* is dispensable for initiating EPI gene expression (2). The evidence on *Oct4* and *Sox2* function suggests that the functional interaction of the NANOG-SOX2-OCT4 trinity comes into force only at the ICM of the blastocyst stage, when lineage specification begins. At this stage, SOX2 and OCT4 play crucial roles in EPI maturation and in the formation of TE and PE, respectively (2, 53, 54).

Besides a substantial fraction of pluripotency factors being conserved in human and murine EPI, including *OCT4*, *SOX2*, *NANOG*, *FGF4*, *TFCP2L1*, and *PRDM14*, human EPI uniquely expresses *KLF17*, *HESX1*, *ELF3*, and *TFAP2C* and *GCM1*, the latter two being described as TE-associated genes in mice (47, 48, 55). In humans, loss of *OCT4* affects the expression of EPI genes such as *NANOG* and *KLF17*, and compromises PE formation (56), fortifying the role of *OCT4* in PE progression as has been reported in mice (12, 50).

## A Roadmap to EPI and PE Establishment

The successful establishment of the EPI and PE requires both cell identity acquisition and physical cell sorting, culminating in the formation of a distinct PE layer (57). The process of cell fate specification within the ICM is characterized by a series of sequential events. Initially, *Nanog* and *Gata6* are activated in early blastomeres, co-expressed in uncommitted progenitors, and eventually become uniquely expressed in EPI-biased and PE-biased cells, respectively. Both *Nanog* and *Gata6* begin to be expressed at the 8-cell stage, gradually accumulating in the ICM cells. At the onset of blastocyst formation (32-cell stage), all ICM cells co-express *Nanog* and *Gata6*. Between the 32-cell and 64-cell stages, their expression becomes mutually exclusive in an asynchronous manner, leading to the emergence of lineage-biased cells in a "salt-and-pepper" mosaic pattern. Therefore, EPI-biased cells exhibit higher expression of *Nanog*, whereas PE-biased cells show higher expression of *Gata6*.

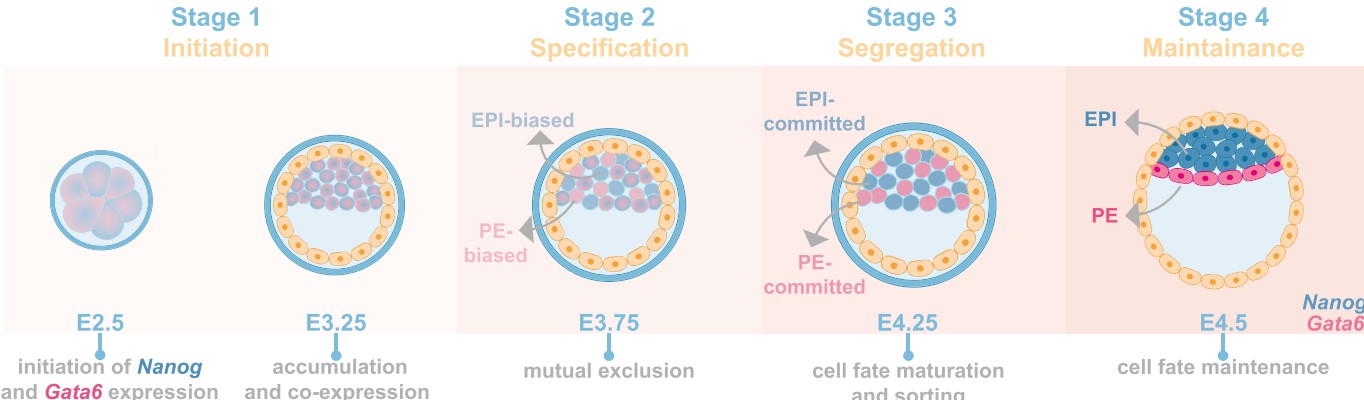

**Figure 2. Four stages of EPI and PE establishment.**
EPI and PE formation is the result of four consecutive stages, based on the expression of the two master regulators of these two layers, *Nanog* and *Gata6*, respectively. Their expression commences at the 8-cell stage, and by E3.25, they are co-expressed by all ICM cells (stage 1). At E3.75, *Nanog* and *Gata6* become mutually excluded, thereby giving rise to EPI- and PE-biased cells (stage 2). By E4.25, *Nanog* and *Gata6* expression is dominant in EPI- and PE-committed cells, and other lineage-specific transcriptional regulators become expressed, facilitating cell fate maturation (stage 3). At this stage, EPI and PE cells physically start sorting and ultimately localize to their respective position, where lineage identity is maintained until post-implantation differentiation cues trigger their reorganization (stage 4). Schematic representation is simplified, and cells are colored as entities. No nucleus is presented at stages 1, 2, and 3. However, all events take place in the nucleus. Icons were obtained from BioRender.com.

We propose four sequential stages for EPI and PE lineage formation based on the expression patterns of *Nanog* and *Gata6* (Fig 2):

1. Stage 1 (E2.5–E3.25): Initiation—This stage encompasses the initial expression of *Nanog* and *Gata6* in single blastomeres and their accumulation until all ICM cells co-express these factors.
2. Stage 2 (E3.25–E3.75): Specification—During this stage, the co-expression state resolves through the mutual exclusion of *Nanog* and *Gata6*, leading to the emergence of lineage-biased progenitors.
3. Stage 3 (E3.75–E4.25): Segregation—At this stage, either *Nanog* or *Gata6* becomes dominantly expressed, and cells mature into EPI- or PE-committed cells, respectively. Lineage maturation coincides with their physical sorting within the ICM.
4. Stage 4 (E4.25 onward): Maintenance—This stage includes the maintenance of EPI and PE identity and positioning until post-implantation differentiation occurs.

## Emergence of EPI and PE transcriptional programs (stage 1)

To comprehend the process of lineage specification, it is essential to first understand how the master regulators of the EPI and PE programs, *Nanog* and *Gata6*, are initially expressed and transcriptionally controlled within the distinct ICM subpopulations. NANOG plays a critical role in initiating EPI specification potentially through the recruitment of key pluripotency network components (SOX2, OCT4, and KLF4) to transcriptionally active sites (58), a process essential for acquiring and maintaining pluripotency in mESCs (58, 59, 60, 61, 62, 63, 64). Although the proposal suggesting NANOG's collaboration with EPI factors in initiating the EPI state offers an explanation for the "salt-and-pepper" distribution during

blastocyst formation, the upstream mechanisms regulating the initial expression of *Nanog* need further exploration.

Epigenetic factors have been proposed as the drivers behind the transcriptional activation of significant EPI lineage markers, such as *Nanog* and *Sox2*. A recent study suggested that the initial expression of *Nanog* and other EPI factors could be regulated by BRD4, a bromodomain and extra-terminal domain (BET) family member (65). In mESCs, BRD4 activates the transcription of core pluripotency genes, such as *Nanog*, *Oct3/4*, and *Prdm14*, by associating with their enhancers (66, 67, 68) (Fig 3). BRD4 is recruited to acetylated lysine residues of H3 and H4 histones and recruits the positive transcription elongation factor b (P-TEFb), which in turn reforms RNA Pol II and other factors required for overcoming the elongation block (69, 70). In this manner, BRD4 facilitates transcript elongation both on protein-coding regions and at enhancers (71). Short-time BRD4 inhibition in 8-cell- and 16-cell-stage morulae resulted in a diminished number of cells positive for NANOG and other pluripotency markers, indicating compromised EPI initiation. However, BRD4 inhibition had a negligible impact on *Gata6* expression, showing that PE initiation is independent of EPI signals, as it has been previously described (72, 73). Similarly, short-time treatment of E3.5 blastocysts with the inhibitor did not affect PE formation. In contrast, long-time BRD4 inhibition in E3.5 blastocysts severely compromised both EPI and PE maintenance, denoting the necessity of EPI signals for PE specification and maintenance (Table 1). Indeed, *Nanog* depletion does not prevent the induction of *Gata6* but impedes the expression of later PE markers, showing that EPI cells are the primary source for PE maturation but not for PE initiation (72, 73, 74).

Surprisingly, *Brd4* null E3.5 blastocysts showed normal EPI formation contrary to their inhibitor-treated counterparts. However, *Nanog* and *Sox2* mRNA levels were reduced, indicating that although BRD4 may not be essential for EPI specification, it is required for the correct expression of EPI markers (Fig 3). The dispensable role of BRD4 in EPI formation in *Brd4*$^{-/-}$ embryos can

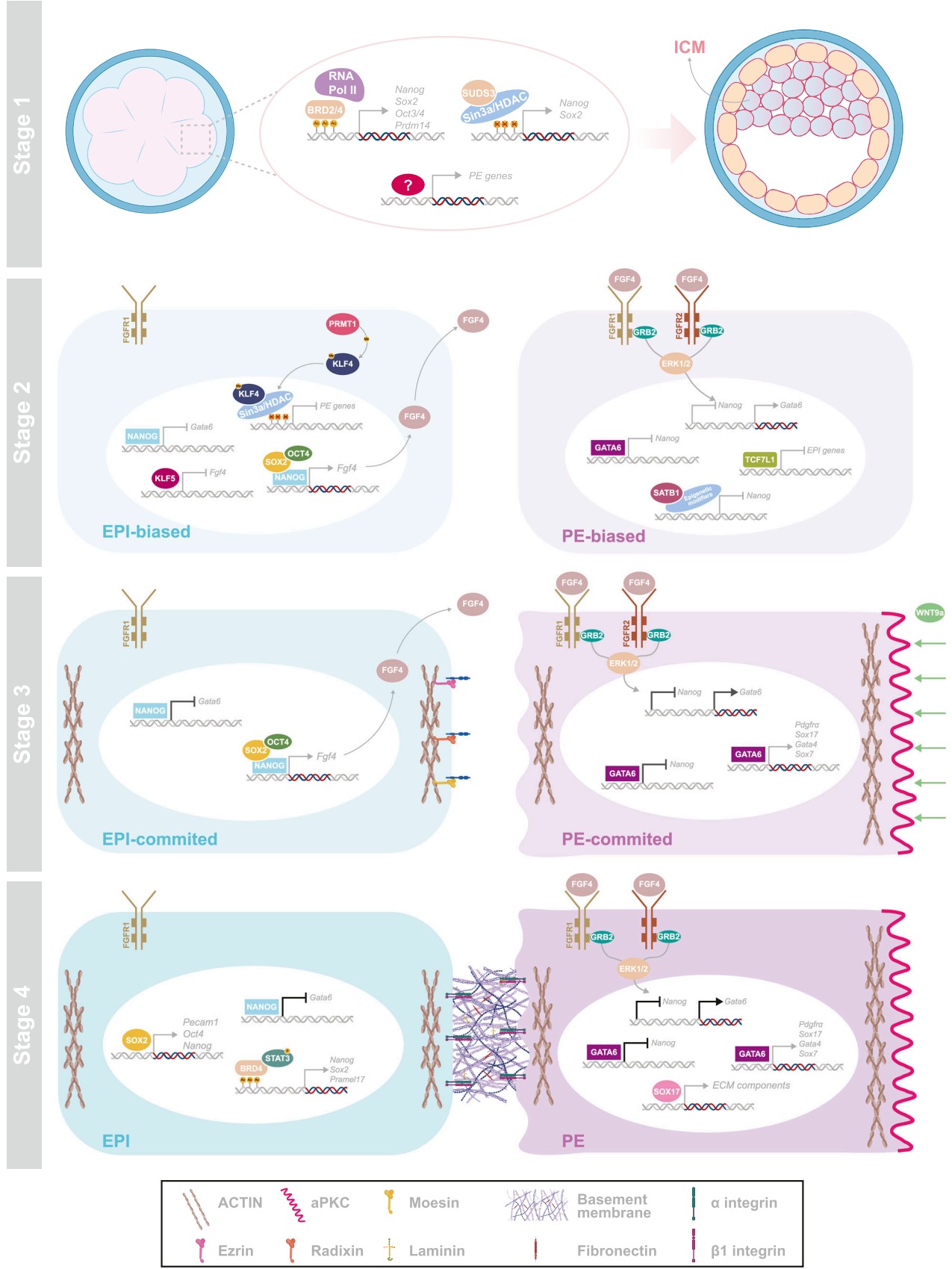

be explained by its possible redundant function with BRD2, another BET family member. Double *Brd2*/*Brd4*-deficient morulae showed diminished NANOG⁺ cells, demonstrating that the complementary tasks of BRD2 and BRD4 are required for correct EPI gene induction and specification (Table 1) (65).

Histone lysine acetylation is a hallmark of transcriptional activation and gene expression, whereas its deacetylation typically represses gene expression by inducing a closed chromatin state (93). However, post-initiation deacetylation caused by histone deacetylases (HDACs) is important for transcriptional elongation and plays an essential role in the active expression of many genes (93). HDACs are found in distinct multiprotein complexes, including the Sin3a complex (94). SUDS3 (or SDS3) is an integral component of the Sin3a/HDAC complex, and *Suds3* deletion leads to embryonic lethality at the peri-implantation stage in mice (95). Although *Suds3*-deficient embryos form TE, they fail to implant, indicating a defective functionality of this layer (78). In the absence of *Suds3*, the percentage of NANOG⁺ and SOX2⁺ cells was significantly reduced at the morula and/or the blastocyst stage, as well as the mRNA levels of *Nanog* and *Sox2*, indicating impaired EPI lineage formation (Table 1). This denotes that SUDS3 is a positive transcriptional regulator of EPI gene expression and might be involved in their initial activation, contributing to the correct EPI specification (Fig 3). In addition, *Suds3*-deficient embryos exhibited significantly elevated *Gata6* mRNA levels and impaired EPI/PE segregation. Despite *Gata6*'s expression, mature PE cells were scarcely found in *Suds3*-deficient blastocysts, suggesting a defect in PE formation, reflecting the phenotype seen in *Nanog*⁻ᐟ⁻ embryos (72, 73, 74) (Table 1). *Suds3*-deficient 8-cell embryos exhibited increased acetylation of lysine 12 on H4 (H4K12ac), a well-known activating chromatin mark (93). Although BRD4 has been shown to regulate master developmental genes associated with pluripotency by binding hyperacetylated histone H4 (96), the excessive H4K12ac observed in *Suds3*-deficient embryos was insufficient to induce BRD4-mediated transcription of key EPI genes. This indicates that the interplay between acetylation and deacetylation could be crucial for transcription elongation during lineage marker activation (93).

Considering that GATA6 is the master regulator of the PE program, the question regarding the mechanism that regulates its initial activation in early blastomeres remains largely unanswered.

Analysis of *Nanog*⁻ᐟ⁻ embryos showed that NANOG does not regulate *Gata6* induction because *Gata6* was activated as expected (72, 73). It has been demonstrated that E3.25 *Klf5*⁻ᐟ⁻ embryos exhibited initial *Nanog* and *Gata6* expression similar to WT embryos. However, later on, lineage specification skewed toward the PE fate, and despite the slight increase of GATA6 protein expression levels in *Klf5*⁻ᐟ⁻ embryos, the *Gata6* mRNA expression was not significantly altered, suggesting that KLF5 does not regulate *Gata6* at the transcriptional level (97).

### EPI and PE lineage specification (stage 2)

Despite the advances in comprehending the mechanisms that regulate early cell fate decisions, many significant questions remain unresolved, for example, how the emergence of the "salt-and-pepper" distribution of lineage progenitors occurs or, in other words, how the resolution of the co-expression stage is regulated at a transcriptional level (stage 2).

It is well described that the mutually exclusive expression of *Nanog* and *Gata6* arises from their ability to repress each other's transcription, based on their respective binding sites identified in chromatin immunoprecipitation studies of mESCs (27) and induced XEN (iXEN) cells (98). NANOG can bind to the *Gata6* promoter in mESCs and suppress its transcription (Fig 3) (27, 38). In addition, *Gata6*⁻ᐟ⁻ embryos lack PE, and ICM cells homogeneously express NANOG, indicating a shift toward the EPI fate. This suggests that the initial expression of *Nanog* is independent of GATA6 (Table 1) (44, 76). Remarkably, NANOG^neg cells were detected at low frequency in *Gata6*⁻ᐟ⁻ blastocysts and were found to express the TE marker CDX2, indicating that in the absence of NANOG and GATA6, the ICM cells located next to the blastocyst cavity may adopt TE fate (44). Conversely, *Nanog*⁻ᐟ⁻ embryos display the pan-ICM expression of GATA6; however, later PE markers are not expressed, indicating the non–cell-autonomous requirement from the EPI cells for PE maturation (Table 1) (74). In summary, it is widely acceptable that the imbalance between *Nanog* and *Gata6* expression is the basis of EPI versus PE specification (Fig 3).

The FGF4 signaling pathway is the most described pathway that influences the reciprocal inhibition of the EPI and PE lineage-specific programs. FGF4 receptors, FGFR1 and FGFR2, belong to the receptor tyrosine kinase (RTK) transmembrane cell-surface

**Figure 3. Overview of the mechanisms involved in EPI and PE fate determination.**
At stage 1, the expression of *Nanog* and *Gata6* is induced at the 8-cell stage and epigenetic regulators, BRD2/4 and SUDS3, regulate the expression of *Nanog* and other EPI factors. *Nanog* and *Gata6* become accumulated and co-expressed in the ICM of the E3.25 blastocyst. At stage 2, the mutual repression of *Nanog* and *Gata6* leads to their exclusive pattern of expression and more mechanisms enhance the emergence of EPI- and PE-biased cells. KLF5 ensures the timely activation of *Fgf4* by suppressing it. The trinity of pluripotency factors NANOG-OCT4-SOX2 positively regulates its expression and activates it in a subset of cells. Methylated KLF4 by PRMT1 recruits the repressive mSin3a/HDAC complex at PE gene promoters and inhibits their expression in the same subset. FGF4 is secreted by the EPI-biased cells and received from the subset of cells, which expresses both FGFR1 and FGFR2. The signal is transduced by the GRB2 adaptor protein, and activated ERK1/2 enters the nucleus and induces *Nanog* suppression and *Gata6* expression. Other factors, such as TCF7L1 and SATB1, suppress EPI-related genes. At stage 3, EPI- and PE-fated cells mature by the activation of other lineage-specific factors into EPI- and PE-committed cells. EPI-committed cells continue providing sustained supply of FGF4 to PE-committed cells, which results in enhanced *Nanog* repression and *Gata6* expression. GATA6 sequentially activates the expression of PE-specific genes. Along with their maturation, PE-committed cells start sorting to form the PE epithelium, a process regulated by cytoskeleton proteins (ACTIN) and cross-linker proteins (ezrin–radixin–moesin), which affect the effective membrane tension of these cells and induce their physical movement. The establishment of apicobasal polarity via aPKC accumulation contributes to their successful sorting. PE-committed cell identity and position might be stabilized by WNT9a, which is accumulated around the blastocoel. At stage 4, EPI and PE fates are irreversibly consolidated and maintained through the enhanced expression of EPI genes by SOX2 and BRD4/pSTAT3 in EPI cells and PE genes by GATA6 in PE cells. SOX17 induces the expression of ECM components that contribute to the formation of a basement membrane, between the PE epithelium and the EPI, supporting lineage layer integrity. Receptor proteins, such as *β*1 integrin, expressed from both EPI and PE cells support the strong cohesion between the two cell types. Icons were obtained from BioRender.com.

**Table 1. Overview of the components involved in the different stages of EPI and PE determination.**

| Gene | Function | Genotype/manipulation | Phenotype regarding EPI/PE |
|---|---|---|---|
| *Nanog* | Transcription factor | $Nanog^{tm1Yam/tm1Yam}$ (38, 74) | Lack of EPI; no mature PE (38, 73, 74) |
| | | $Nanog^{gt1/gt1}$ and $Nanog^{gt2/gt2}$ (73, 75) | |
| *Gata6* | Transcription factor | $Gata6^{tm2.2Sad/tm2.2Sad}$ (44, 76, 77) | Lack of PE (44, 76) |
| | | $Gata6^{tm2.2Sad/+}$ (44, 76, 77) | Decreased PE specification; lineage distribution switches to EPI (44, 76) |
| *Brd4* | Epigenetic factor | Chemical inhibition (65) | EPI defect; EPI gene expression compromised (65) |
| | | $Brd4^{lacZ/lacZ}$ (65) | |
| *Brd2* | Epigenetic factor | $Brd2^{tg/tg}$ (65) | Normal EPI development (65) |
| *Brd2/Brd4* | | $Brd2^{tg/tg}$; $Brd4^{CRISPR-/-}$ (65) | EPI defect (65) |
| *Suds3* | Epigenetic factor | $Suds3$-dsRNA (78) | EPI defect; EPI gene expression compromised; PE defect (78) |
| *Grb2* | Adaptor protein | $Grb2^{\Delta}$ (79, 80) | Lack of PE (79) |
| *Tcf7l1* | Transcription factor | $Tcf7l1^{CRISPR-/-}$ (81) | Decreased PE specification; lineage distribution switches to EPI (81) |
| *Satb1* | Epigenetic factor | $Satb1$-siRNA (82) | Decreased PE specification; lineage distribution switches to EPI (82) |
| | | $Satb1$-mRNA (82) | Decreased EPI specification; lineage distribution switches to PE (82) |
| *Prmt1* | Epigenetic factor | Chemical inhibition (83) | Decreased EPI specification; lineage distribution switches to PE (83) |
| *Fgf4* | Ligand | $Fgf4^{\Delta2,3}$ (84, 85) | Lack of PE (84, 85) |
| *Fgfr1* | Receptor | $Fgfr1^{\Delta8-14}$ (84) | Decreased PE specification; lineage distribution switches to EPI (84) |
| *Fgfr2* | Receptor | $Fgfr2^{\Delta}$ (84) | Delayed PE specification (84) |
| *Fgfr1/Fgfr2* | | $Fgfr1^{\Delta8-14}$; $Fgfr2^{\Delta}$ (84) | Lack of PE (84) |
| *aPKC* | Kinase | Chemical inhibition (86) | Decreased PE maturation; failed PE sorting (86) |
| *Sox2* | Transcription factor | $Sox2^{\beta geo/\beta geo}$ (87) | Lack of EPI at post-implantation (87) |
| | | $Sox2^{tm1.1Lan/tm1.1Lan}$ (2, 88) | Decreased PE specification; increased uncommitted ICM cells (2) |
| *Sox17* | Transcription factor | $Sox17^{fl/fl}$ (46, 89) | PE premature differentiation; disorganized PE epithelium (46) |
| *Stat3* | Transcription factor | $Stat3^{fl/fl}$ (90) | Lack of EPI and PE (90) |
| *Intb1* | Receptor | $Intb1^{tm1Efu/J}$ (91, 92) | Disorganized PE epithelium (91) |

receptors and similarly transduce the FGF4 signal through GRB2, which results in ERK phosphorylation and ultimately in the regulation of target gene transcription (99). Inhibition of RTK signaling via FGFR inhibition at the onset of *Gata6* expression (E2.5) resulted in no *Gata6*-expressing cells in the ICM (74). Similar results were observed in $Grb2^{-/-}$ blastocysts, where *Gata6* expression was absent, and the ICM uniformly expressed *Nanog* (79). However, the effects of chemical inhibition or genetic ablation were only investigated at late stages when lineages are specified or already segregated. Later studies established FGF signaling as the primary pathway that governs PE restriction and maintenance but not initiation. Specifically, $Fgf4^{-/-}$ embryos do not form PE, as indicated by the lack of early (*Gata6*) and late PE-specific genes (*Gata4* and *Sox17*) at E4.5 (Table 1) (84). However, *Fgf4* mutant embryos express *Gata6* until E3.25, although it cannot be maintained afterward,

leading to an exclusively comprised NANOG$^{+}$ ICM. Hence, FGF4 signaling does not regulate the initiation of *Gata6* expression and, as a result, the PE transcriptional program, but it is indispensable for the establishment of the "salt-and-pepper" distribution and the maturation of both lineages.

*Fgf4* expression is restricted to the ICM at E3.5 (100). Bimodal *Fgf4* expression within the ICM cells precedes the exclusive expression of *Nanog* and *Gata6* (3, 101). Subsequently, a subset of cells expresses *Fgfr2*, which becomes reciprocal to the *Fgf4* expression. *Fgf4*-expressing cells are regarded as EPI-biased, and sustained FGF4 supply to neighboring *Fgfr2*-expressing cells drives PE specification (Fig 3) (29, 44, 72, 84, 101, 102). FGFR1 is uniformly expressed in all ICM cells earlier than *Fgfr2* (E3.25 versus E3.5) (99). The updated model suggests that FGF4 binding on *Fgfr1*-expressing cells stimulates low ERK activity, facilitating *Nanog* level

maintenance and EPI identity acquisition in a subset of cells. Afterwards, the supposed PE cell subset initiates expressing *Fgfr2*, where the FGF4 signal is transduced through both FGFR1 and FGFR2, resulting in robust ERK activation, which leads to diminished *Nanog* expression and the complete suppression of the EPI program. Therefore, *Gata6* expression is enhanced and promotes the activation of the PE program.

This mechanism suggests the establishment of lineage-biased cells from a common signal, which elicits opposite effects based on the availability of the signal and the intercellular differences between FGF target and receptor expression. Initially, the process depends on the expression of *Fgf4* by a group of cells and, later, on the expression of *Fgfr2* by the PE-fated subpopulation. It has been proposed that *Fgf4* expression is activated by the stochastic coordinated expression of *Nanog* and other EPI markers in a group of cells at the 16-cell stage (103). At E3.5 (mid-stage 2), the co-expression of these factors enhances *Fgf4* expression for proper EPI specification in the ICM, facilitating the PE specification of neighboring cells (72, 73, 74, 103, 104) (Fig 3). Of note, the initial factors that trigger *Fgfr2* expression remain unidentified. It was demonstrated that KLF5 occupies the *Fgf4* locus and directly represses its expression between E3.0 and E3.25, potentially explaining how bimodal *Fgf4* expression is generated (97). In this interval, *Fgf4* expression is highly induced in a subset of cells in the ICM of $Klf5^{-/-}$ embryos, and late markers of PE are prematurely expressed, indicating KLF5's role in suppressing precocious activation of the FGF4-FGFR-ERK pathway, securing EPI development (Fig 3).

It was previously postulated that ERK signaling does not play a role in human PE formation because ERK inhibition in different staged embryos did not affect EPI or PE cells in late human blastocysts (50, 105). However, recent studies, where both FGFR and ERK were inhibited from the EPI/PE branchpoint (D5) onward, resulted in the almost complete absence of PE in favor of EPI cell number (21 *Preprint*, 106 *Preprint*). Consistently, both FGF2 and FGF4–heparin increased the PE population at the expense of the EPI, fortifying the theory that FGF signaling may be involved in human EPI/PE commitment, as has been described in mice.

Even though it was considered that the bimodal expression of *Fgf4* within the early mouse ICM could be the first sign of distinction within the population, $Fgf4^{-/-}$ ICM cells maintain a consistent level of gene expression variability, indicating that *Fgf4* is dispensable for generating the initial molecular heterogeneity (101). It was shown that primary ICM heterogeneity emerges between the 16-cell and 32-cell stages because of the coordinated expression of EPI markers in a subset of cells in both mouse and human ICM progenitors (103). $Nanog^{-/-}$; $Gata6^{-/-}$ dKO mouse embryos lack mature EPI and PE lineages because ICM cells remain in a progenitor state. However, dKO embryos exhibit ICM heterogeneity shown by cell-to-cell variability in EPI factors, indicating that the initial ICM heterogeneity exists before EPI and PE differentiation, and this could be the basis for the onset of ICM differentiation (103). In agreement, primary ICM heterogeneity based on correlation analysis of FGF4 and NANOG expression has been described in human blastocysts; hence, unspecified progenitor cells were distinguished from differentiating/-ed cells. The presence of a discrete population that is transcriptionally distinct from EPI and PE would be in line with the notion of a distinct ICM signature (55, 103).

A recent study provided insight into how this heterogeneity arises in mice by demonstrating that during the co-expression stage, NANOG and GATA6 co-bind at both EPI and PE cis-regulatory elements (CREs) in vitro and in blastocysts, though with different affinities (45). This considerable overlap in binding at CREs that regulate the EPI and PE transcriptional networks contributes to maintaining ICM cells in a poised state and enables rapid bifurcation of unbiased progenitor cells into divergent lineages. EPI TFs were redirected from EPI to PE CREs, when *Gata6* expression surpassed the one of *Nanog* in vitro, suggesting a potential mechanism through which GATA6 facilitates the rapid activation of the PE network (45).

Although the FGF4/ERK regulatory axis continues to serve as the predominant force of lineage specification, other signaling cascades and epigenetic factors have also been identified to play a critical role in the process of EPI/PE specification. Despite the expression of Wnt agonists, antagonists, and components involved in the transduction machinery in the mouse pre-implantation embryo (107, 108), the Wnt/$\beta$-catenin pathway is considered dispensable for pre-implantation development because $Ctnnb1^{-/-}$ embryos develop normally until gastrulation (109). Recently, the involvement of Wnt/$\beta$-catenin in cell fate acquisition within the ICM has been acknowledged (81). Wnt signaling inhibition resulted in the enrichment of the PE population at the expense of the EPI, indicating that Wnt signaling levels are essential for the balanced EPI/PE specification. Interestingly, Wnt modulation did not affect TE formation, suggesting that canonical Wnt signaling is tightly associated with the development of divergent cell fates in the ICM. TCF/LEF family of TF regulates gene expression downstream of the Wnt/$\beta$-catenin cascade. A role of TCF7L1, a TCF/LEF family member and non–lineage-specific TF, was demonstrated in the establishment of the initial heterogeneity in the ICM and as a potential candidate for driving PE specification. TCF7L1 is a well-described repressor of pluripotency gene expression in mESCs (110, 111), and deletion of *Tcf7l1* is embryonic lethal (112). However, its involvement in pre-implantation lineage decision was not previously described. $Tcf7l1^{-/-}$ embryos exhibited a significantly reduced number of PE cells and higher number of EPI cells, indicating that *Tcf7l1* depletion promotes EPI specification at the expense of the PE (Table 1). In vitro study confirmed the indispensable role of TCF7L1 in PE specification because $Tcf7l1^{-/-}$ mESCs were impaired to differentiate toward PE. The overexpression of *Tcf7l1* activated PE gene transcription and induced differentiation of mESCs into PE-like cells. Interestingly, binding of TCF7L1 on PE gene promoters was not detected, indicating that TCF7L1 does not actively promote PE fate. Conversely, it was demonstrated that TCF7L1 binds and represses genes related to EPI progression, thus allowing the PE program to take over (81) (Fig 3).

PRMT1, a protein arginine methyltransferase, was shown to methylate KLF4 in mESCs, which in turn binds and recruits the repressive mSin3a/HDAC complex at PE gene promoters (83) (Fig 3). As a result, the expression of PE genes such as *Gata4* and *Gata6* is inhibited, and their mRNA levels are reduced. Depletion of *Prmt1* induced the emergence of a small population resembling the XEN cells, as identified by scRNA-seq. *Prmt1* depletion reduces the recruitment of the mSin3a/HDAC complex, and this causes the hyperacetylation of H3 and H4 in the promoters of the PE genes,

which leads to an open chromatin state and PE gene activation. Inhibition of PRMT1 in mouse embryos resulted in an increased percentage of GATA6+ cells (Table 1). In addition, mESCs treated with the same inhibitor, before being injected into blastocysts, integrated into both EPI and PE, indicating the PE bias of these cells compared with non-treated cells. Collectively, PRMT1 likely has a role in the "salt-and-pepper" pattern emergence by restricting the PE program induction. Although this described epigenetic mechanism could help us close some of our current gaps in knowledge, whether this mechanism promotes divergent fates in vivo remains to be determined.

SATB1, a chromatin modifier that provides docking sites for other chromatin remodeling proteins (113), was shown to have a critical function on cell fate commitment in the mouse pre-implantation embryo (82). Satb1 knockdown (KD) in mouse zygotes resulted in a significant reduction of PE cells and an increase in EPI cells in the early blastocyst (Table 1). Accordingly, the overexpression of Satb1 caused the opposite effect with ICM containing significantly increased number of PE cells and significantly lower number of EPI cells. Thus, SATB1 appears to have a distinct role in the differentiation of the ICM into PE or EPI. Analysis of Satb1 KD embryos at the 16-cell stage and at the early blastocyst, the 32-cell stage, showed that there was no effect on the number of EPI and PE cells, indicating that SATB1 is dispensable for the initiation of the PE program. Satb1 is upregulated in the PE precursors, suggesting that it has a key function in PE commitment and its divergence from the EPI (82). SATB1 was shown to repress Nanog in mESCs (114), and the overexpression of Satb1 in embryos resulted in a significant decrease in Nanog mRNA and drove ICM cells to preferentially differentiate into PE rather than EPI (Fig 3). Therefore, SATB1 is described as a new epigenetic regulator that controls cell fate choice in the pre-implantation ICM.

**EPI and PE lineage segregation and maturation (stage 3)**

After the exclusive expression of Nanog and Gata6 in EPI and PE precursors, respectively, lineage-specific gene activation leads to mature identity acquisition, coinciding with their physical segregation into two separate compartments through sorting (stage 3). Concurrently, with the sequential activation of PE-specific TFs by GATA6, PE precursors migrate to the ICM surface and form a single-layered mature epithelium, which encloses the EPI between PE and TE (115).

The role of positional induction of fate in lineage segregation, where gene expression depends on a cell's position within the ICM, has been a topic of debate. Computational simulation of the survival, movements, and gene expression during the progression of lineage segregation showed that the combination of the cell sorting model, where cells sort according to their gene expression patterns, and the induction model, where gene expression state is position-dependent, gave the best score for the observed data (116). These simulations substantiate that the conjunction of cellular movements and positional induction might be essential for EPI and PE segregation. The theory that positional information may provide instructive cues that influence differential gene expression was supported when ICM cells facing the cavity from E3.5 onward predominantly correspond to

the presumptive PE group at E4.5 (101). In agreement, live imaging of Pdgfra^H2B-GFP blastocysts showed that once PDGFRα+ cells reached the cavity, they rarely changed their position, suggesting that other mechanisms are in effect to facilitate position safeguarding (57).

WNT9A, whose expression is specifically detected in the cells surrounding the blastocoel cavity, might be a potential regulator of gene expression stabilization from specification to segregation (116) (Fig 3). Although the overexpression of Wnt9a and Gata6 alone was insufficient to elicit a significant effect on the cells' location, the combination of Gata6 and Wnt9a overexpression propelled cells to the surface of the ICM.

Physical cell sorting within the ICM results from active cell movements, adhesion, and selective apoptosis (57, 116). It has been shown that ACTIN and ACTIN cross-linkers, as part of the cytoskeleton, play a substantial role in orchestrating cell movements and coordinating cell sorting (116). Overall, although the segregation of the two lineages involves active movements, positional signals appear to affect PE establishment, which is intimately associated with its maturation (Table 1).

Cell polarity is another aspect that can affect fate stabilization. Polarization of PE cells has been demonstrated to be essential for their final fate acquisition and the sorting out of the PE and the EPI (86, 117, 118, 119). Atypical protein kinase C (aPKC), known for its role in establishing apicobasal polarity and cell fate across multicellular organisms, is indispensable for the segregation of PE and EPI cells, as in the absence of aPKC, PE cells fail to sort successfully (86) (Fig 3). Initially, aPKC is homogeneously expressed throughout the embryo and later becomes enriched in PE precursors accompanying their specification. aPKC polarization in PE cells is a hallmark of their epithelialization. Exogenous inhibition of aPKC in the segregation window showed that although PE cells change positions within the ICM, they cannot maintain it to form an epithelial monolayer, whereas aPKC inhibition after the PE layer is formed did not affect PE integrity (Table 1). Thus, aPKC activity is required for the organization of the PE epithelium; however, it is dispensable for its maintenance. In addition, aPKC inhibition caused retention of GATA6 in supposedly mature PE cells, without the expression of later PE genes, suggesting aPKC-dependent impaired PE maturation.

Recent studies highlighted the mechanical processes underpinning robust PE segregation from EPI (120). PE precursors display a more uneven surface morphology than their EPI counterparts because of their high amplitude of surface fluctuations caused by the variability of the effective membrane tension on their cell surface, regulated by the ezrin–radixin–moesin (ERM) protein family. Triple knockdown of ERM (siERM) resulted in decreased effective membrane tension, thus in high surface fluctuations, and siERM cells sorted toward the outside of ES aggregates, indicating that cell sorting is regulated by differences in surface fluctuations. Similarly, the injection of siERM cells in 8-cell morula and blastocysts resulted in the integration of a considerable fraction in the presumed PE layer, denoting that cell-surface fluctuations induce physical sorting in the ICM. Although EPI and PE precursors display similar size, PE precursors exhibited a more fluid cell shape than the EPI cells, which was associated with the high surface fluctuations. The combination of high cell fluidity and surface

fluctuations appears to be the driving force for the demixing of PE cells from the EPI population (Fig 3).

Despite the establishment of the mutually exclusive expression of EPI and PE TFs by E3.75, ultimate cell fates are not yet consolidated as they can be still altered through modulation of the FGF signaling levels by E4.5 (121, 122). Specifically, PE precursors present high plasticity by this stage, and they possess competence to generate EPI and TE when released from FGF4 stimulation, with a preference for the latter (123, 124). It has been suggested that early PE maintains the capacity to regenerate all the cell types of the blastocyst through the activity of the JAK/STAT signaling, which supports bivalent enhancer activation related to the EPI and TE fate (123). Therefore, sustained exposure to FGF signaling is required for PE maturation of PE precursors, and inhibition of the FGF cascade is necessary to block the PE program in EPI precursors, allowing the co-expression of EPI-associated factors (84, 121). Later, during blastocyst implantation, FGF4 induces *Nanog* down-regulation in EPI cells in a cell-autonomous manner through FGFR1, indicative of EPI lineage maturation, suggesting that FGF signaling is required for the EPI transition to a primed pluripotent state (84, 99, 125).

Besides the role of FGF signaling in lineage maturation, explained in the previous section, FGF/ERK has also been indirectly implicated in the EPI/PE sorting process. Embryos treated with ERK1/2 inhibitor from the 8-cell stage not only prevented PE specification (86), but they also presented low levels of aPKC in the ICM and barely any polarized cells (126). On the contrary, in embryos treated with FGF4, ICM was entirely comprised of PE cells, yet aPKC polarized only in cells that were in contact with the cavity, indicating that aPKC polarization is cell position–dependent. These findings suggest that aPKC levels and polarization rely on FGF signaling within the ICM. In addition, surface fluctuations appear to be driven by FGF signaling because FGF treatment of isolated ICM cells resulted in higher surface fluctuations, whereas inhibition of ERK attenuated surface fluctuations. Nonetheless, the involvement of FGF signaling with molecular and mechanical mechanisms that propel the sorting of EPI and PE lineages is possibly a consequence of PE identity acquisition downstream of FGF signaling rather than its direct association with these processes.

## EPI and PE maintenance (stage 4)

Lineage allocation and maturation are milestones for blastocyst's competence for implantation and post-implantation development. Hence, maintaining the specified lineages within the ICM during the peri-implantation window is essential to ensure proper morphogenetic transformation shortly after implantation. Yet, the mechanisms that may contribute to EPI maintenance and PE epithelial integrity remain largely unknown.

SOX2 is one of the earliest markers of ICM progenitors, but its role in regulating cell fate only manifests at the late blastocyst stage. Although EPI cells are absent in *Sox2* null post-implantation embryos (87), EPI progression appears unaffected until E3.75 (54). However, the expression of EPI markers such as *Pecam1*, *Oct4*, and *Nanog* was severely reduced or undetectable already at E4.25. The EPI cell number was significantly reduced at later developmental stages. In addition, diapause-induced *Sox2*$^{-/-}$ embryos exhibited significantly reduced EPI cells at later developmental stages (2)

(Table 1). Of note, the number of PE cells remained unchanged across the prolonged pre-implantation stages. These observations suggest that even though SOX2 is dispensable for EPI initiation, it is required for maintaining EPI identity at E4.25 onward (Fig 3).

The formation of a mature PE epithelium includes the polarization of PE cells, which posits their apical side adjacent to the cavity and their basal side to the basement membrane, lying between the PE and EPI. The basement membrane is composed of extracellular matrix (ECM) components, including collagen IV, laminin, vitronectin, and fibronectin (127, 128, 129). β1 integrin (*Intb1*) is expressed in both EPI and PE (130), and heterodimers of β1 integrin with α integrins can bind multiple components of the basement membrane. Although β1 integrin is not required for EPI or PE specification, *Intb1*$^{-/-}$ blastocysts exhibit a disorganized PE epithelium and no distinct basement membrane at E5.5, leading to early embryonic lethality. In vitro, PE differentiation assays show that *Intb1*$^{-/-}$ cells normally specify toward PE and establish apical polarity (91). However, *Intb1*$^{-/-}$ cells fail to maintain an intact surface epithelial layer, indicating that β1 integrin is indispensable for maintaining the PE monolayer structure (Fig 3). Presumably, upon PE cell sorting to the surface, PE cells secrete basement membrane components, which are deposited at the interface with the EPI layer and the strong cohesion of EPI and PE is mediated by their mutual binding to the common basement membrane in a β1 integrin–dependent manner.

SOX17, a PE-specific marker, was also found to play a role in sustaining the epithelial integrity of the PE layer. It was shown that E4.5 *Sox17*$^{-/-}$ embryos expressed both EPI and late PE markers, and they exhibited PE cell numbers comparable to WT embryos. In addition, both lineages successfully sorted to their corresponding tissue layer. However, it was observed that in implantation-delayed *Sox17*$^{-/-}$ blastocysts, PE epithelium was abnormal, and PE cells were prematurely migrating along the mural TE (46). These findings indicate that although *Sox17* is not necessary for the specification of the PE, it significantly contributes to maintaining its epithelial integrity (Table 1). SOX17 binds the promoter of genes encoding ECM components, such as laminin and more (131, 132, 133) (Fig 3). Laminin appeared mislocated and homogeneously expressed around PE cells in implantation-delayed *Sox17*$^{-/-}$ embryos, and not at their basal side as expected, indicating that SOX17 may regulate the production of basement membrane components from PE cells, thus controlling the integrity of PE epithelium.

Self-renewal and maintenance of naive pluripotency of mESCs require medium supplementation with LIF, which binds to the LIF receptor complex (LIFR and gp130), and this induces a cascade of phosphorylation. Phosphorylation of STAT3 (signal transducer and activator of transcription 3) leads to its translocation to the nucleus where it regulates target gene transcription (134). Despite the role of STAT3 in maintaining pluripotency in vitro, *Stat3*$^{-/-}$ embryos do not die before E6.5 (135). Phosphorylated STAT3 (pSTAT3) is present from the zygote stage but becomes functional only at the 4-cell stage onward (90). *Stat3*$^{-/-}$ E3.5 blastocysts exhibit normal development, and TE/ICM specification was successfully established. However, at E4.0, *Stat3*$^{-/-}$ blastocysts begin having abnormal phenotypes with variable lineage compositions, with the most common being the reduction of both NANOG$^+$ and GATA4$^+$ cells. By E4.5, *Stat3*$^{-/-}$ blastocysts exhibit dramatic loss of EPI cells and profound reduction of PE cells, whereas TE appears intact (Table 1). Therefore,

STAT3 is required for ICM lineage maintenance after segregation, primarily by regulating EPI maintenance, which in turn affects PE maturation and maintenance (84, 121).

A recent study suggested that STAT3 may partially exert its transcriptional effects and play its role in EPI/PE maintenance in collaboration with BRD4 (65). *Brd4*-deficient E4.25 blastocysts showed a reduced number of EPI and PE cells, with PE cells having a disorganized arrangement rather than forming a single epithelial layer. This places BRD4 as an important factor in the regulation of EPI and PE maintenance. BRD4 inhibition in E3.5 blastocysts considerably reduced the levels of pSTAT3, whereas it resulted in the downregulation of multiple pluripotency genes, as it has been described before in vitro (96). Several of those, including *Nanog*, *Sox2*, and *Pramel7*, were also found downregulated upon STAT3 inhibition, indicating that STAT3 might cooperate with BET proteins, such as BRD4, to regulate EPI maintenance in the implanting embryo (Fig 3).

# Conclusions and Perspectives

The EPI and PE cell fate specification journey begins from apparently identical progenitors located in the ICM of the E3.25 blastocyst and is completed at E4.5, when the two cell lineages are molecularly and morphologically distinct. However, fundamental questions regarding this timeframe remain open. How do bipotent single cells take on separate fates? When and how does the divergence occur? What are the molecular mechanisms underlying the differential transcriptional patterns that dictate how cells make fate decisions?

One of the numerous possible answers to these questions lies in the emergence of the lineage-specific transcriptional programs, which subsequently regulate diverse cell fate acquisition. Signaling pathways, epigenetic regulators, and TFs form interconnected networks that progressively reinforce or undermine certain transcriptional routes. In this review, we have assembled the molecular mechanisms that orchestrate the differential patterns of expression from the early morula until the late blastocyst stage, which might consequently affect the cell fate decision-making process and the determination of the two ICM-derived lineages. The plethora of factors and cascades implicated in this process, as shown by the adverse effects of their deregulated expression, reveals the complexity of the EPI versus PE binary cell fate decision. Elucidating the missing links, which may connect seemingly independent pieces, will offer invaluable insights into the molecular avenues that progenitor cells follow to acquire lineage identity during EPI versus PE specification. Functional, single-cell, and chromatin accessibility–level studies will contribute to tackling currently unmet challenges and deepen our understanding of the complex circuit governing mouse pre-implantation development.

# Acknowledgements

The authors would like to extend their gratitude to the Research Foundation—Flanders for the Ph.D. fellowships awarded to P Athanasouli (11M7822N) and T Vanhessche (1171725N). The Lluis laboratory is funded by the FWO Research Project Grants G091521N and G073622N (to F Lluis), and the C1 KU Leuven internal grant C14/21/115 (to F Lluis).

## Author Contributions

P Athanasouli: conceptualization, investigation, visualization, methodology, and writing—original draft, review, and editing.
T Vanhessche: investigation and writing—original draft.
F Lluis: supervision, methodology, and writing—original draft, review, and editing.

## Conflict of Interest Statement

The authors declare that they have no conflict of interest.

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
