## [Reviewer comments · Life Science Alliance]

Life Science Alliance

Divergent destinies: Insights into the molecular mechanisms underlying EPI and PE fate determination

Paraskevi Athanasouli, Tijs Vanhessche, and Frederic Lluis

DOI: <https://doi.org/10.26508/lsa.202403091>

Corresponding author(s): Paraskevi Athanasouli, KU Leuven and Frederic Lluis, KU Leuven

Review Timeline:

Submission Date:	2024-10-14
Editorial Decision:	2024-11-15
Revision Received:	2024-12-16
Editorial Decision:	2024-12-19
Revision Received:	2024-12-21
Accepted:	2024-12-23

Transaction Report:

November 15, 2024

Re: Life Science Alliance manuscript #LSA-2024-03091-T

Dr. Paraskevi Athanasouli
KU Leuven
Herestraat 49
Vlaams-Brabant 3000
Belgium

Dear Dr. Athanasouli,

Thank you for submitting your manuscript entitled "Divergent destinies: Insights into the molecular mechanisms underlying EPI and PE fate determination" to Life Science Alliance. The manuscript was assessed by expert reviewers, whose comments are appended to this letter. We invite you to submit a revised manuscript addressing the Reviewer comments.

Thank you for this interesting contribution to Life Science Alliance. We are looking forward to receiving your revised manuscript.

Sincerely,

B. MANUSCRIPT ORGANIZATION AND FORMATTING:

Reviewer #1 (Comments to the Authors (Required)):

This comprehensive review covers our current knowledge of the second cell fate decision in mammalian embryos. While the mechanisms that the second cell fate decision have been covered in multiple reviews, this review is unique in that it is entirely focused on this particular fate decision. As a result, the review is very thorough and presents a detailed mechanistic view. Figure 3 and Table 1 are particularly useful, as they comprehensively present all the information. Having said this, I have several comments that could improve the review:

1. Lines 16-18: the authors argue that there is very little known about the mechanisms that control cell fate decisions, but I would argue that as exemplified by this review there is quite a lot known.
2. Lines 37-38: physical cues should also be mentioned in this context, as exemplified by Yanagida et al, Cell, 2022.
3. Line 82: the EPI will form the extra-embryonic mesoderm and the amnion.
4. Line 98: the EPI and PE in a dish section is perhaps the only section that is missing key important references. For example, Ohinata et al, Science, 2022 grew primitive endoderm-like cells in vitro which better capture the functionality of this tissue. Anderson et al, Nature Cell Biology, 2017 should also be discussed, as it reports an alternative culture medium to grow primitive endoderm cells. In the context of human primitive endoderm cells, Linneberg-Agerholm et al, Development, 2019; Mackinlay et al, eLife, 202; and Okubo et al, Nature, 2024 should be discussed.
5. Lines 108-109: the term naïve is restricted to cells cultured in 2i-LIF. Cells cultured in LIF do not represent a naïve pluripotent state.
6. Line 110: in the context of the in vivo counterpart of naïve ESCs Boroviak et al, Nature Cell Biology, 2014 needs to be mentioned.
7. Line 112: there is no ICM at E4.5.
8. Lines 122-123: it would be important to specify that XEN cells have an increased propensity to contribute to the parietal endoderm.
9. Lines 146-147: Otx2 is also expressed in the visceral endoderm of the post-implantation mouse embryo (Acampora et al, Development, 2013).
10. Lines 299-300: reference 78 needs to be more clearly explained, especially in the context of Silva et al, Cell, 2009. While GATA6 is expressed, other PE markers are not, and this needs to be mentioned.
11. Lines 308-318: this section is a bit confusing: first it is stated that FGF signaling is required for the initial GATA6 expression, and then that FGF is not needed for the initiation of PE fate but rather for maintenance. In this section, the concept that Nanog induces FGF4 expression and the phenotype of NANOG null ICMs could be presented.
12. Lines 327-329: here the authors explain that Fgfr1 expression in EPI progenitors maintains low Erk activity and Nanog maintenance, but in the next section they mention that Fgfr1 activation leads to Nanog downregulation. This is a bit confusing. What determines the two different outcomes?
13. Line 338: what triggers Fgfr2 expression in the PE-biased population? If unknown it would be important to flag it.
14. Line 347: the human data appears in the middle of the mouse data. I would suggest making a specific section about human PE.
15. Line 460: this sentence is not clear. Does it mean that WNT9A localization is restricted to the blastocoel?
16. Lines 465-466: reference 109 is not properly discussed here. This reference shows that cell-cell adhesion, migration, and surface tension do not explain sorting. I would remove the reference from here as this work is properly discussed in another section of the review.
17. Lines 515-519: It is not clear why the authors link the Erk inhibition phenotype to a sorting defect. From what is described it seems Fgf is needed to trigger aPKC expression.
18. Line 582: Stat3 KO embryos, not mice.
19. Figure 1: the key differences between mouse and human embryos could be highlighted. The ICM, EPI, and PE have far too many cells. The drawings do not faithfully represent a blastocyst. This also applies to Figure 2.
20. Figure 3:
 - Stage 1: a question mark before PE genes could be added to emphasize that the initial trigger of PE genes is unknown.
 - Stage 2: the authors could include TCF7L1 in the model and draw FGFR1 in EPI cells. Why are late PE genes not expressed at stage 2 if GATA6 is already there? Is it known whether the binding pattern of GATA6 changes at different stages?

Reviewer #2 (Comments to the Authors (Required)):

This is a timely and well written review on Pre vs Epi fate specification. It is very well written, reads very smoothly and is up to date. Importantly, it covers all relevant papers in a fair and accurate manner and well integrates them into coherent summaries and perspectives, that are also well reflected in the figures which are of high quality.

I did not see any mistakes and don't have any meaningful suggestions to improve this timely review.

Point-by-point answer to reviewers: LSA-2024-03091-T

Reviewers' comments:

Reviewer #1 (Comments to the Authors (Required)):

This comprehensive review covers our current knowledge of the second cell fate decision in mammalian embryos. While the mechanisms that the second cell fate decision have been covered in multiple reviews, this review is unique in that it is entirely focused on this particular fate decision. As a result, the review is very thorough and presents a detailed mechanistic view. Figure 3 and Table 1 are particularly useful, as they comprehensively present all the information. Having said this, I have several comments that could improve the review:

We thank the reviewer for recognizing the novelty and significance of our work in the developmental field. We greatly appreciate the constructive comments provided, which aim to enhance the quality of our study. We are truly grateful for the reviewer's thorough and insightful evaluation.

1. Lines 16-18: the authors argue that there is very little known about the mechanisms that control cell fate decisions, but I would argue that as exemplified by this review there is quite a lot known.

We thank the reviewer for this remark, and we have changed the text accordingly.

Page 1 line 16 of the newly submitted manuscript:

Despite progress in understanding the molecular mechanisms underlying the differential transcriptional patterns that dictate how apparently identical cells make fate decisions and how lineage integrity is maintained, a detailed overview of these mechanisms is still lacking.

2. Lines 37-38: physical cues should also be mentioned in this context, as exemplified by Yanagida et al, Cell, 2022.

Page 2 line 36 of the newly submitted manuscript:

Understanding the molecular links between signal transduction, *physical cues*, and gene expression that dictate cell fate allocation is crucial in elucidating the molecular mechanism involved in lineage establishment.

3. Line 82: the EPI will form the extra-embryonic mesoderm and the amnion.

Page 4 line 86 of the newly submitted manuscript:

As development proceeds, the EPI will form most of the embryo proper, along with the extra-embryonic mesoderm *and the amnion*.

4. Line 98: the EPI and PE in a dish section is perhaps the only section that is missing key important references. For example, Ohinata et al, Science, 2022 grew primitive endoderm-like cells in vitro which better capture the functionality of this tissue. Anderson et al, Nature Cell Biology, 2017 should also be discussed, as it reports an alternative culture medium to grow primitive endoderm cells. In the context of human primitive endoderm cells, Linneberg-Agerholm et al, Development, 2019; Mackinlay et al, eLife, 202; and Okubo et al, Nature, 2024 should be discussed.

We thank the reviewer for pointing out that these crucial references were missing from our manuscript, and we have revised the section “EPI and PE in a dish: in vitro models” and we included the following part.

Our manuscript uses the mouse model to describe the EPI vs. PE specification process in-depth. While a dedicated section on human primitive endoderm development would be valuable, it would significantly increase the article’s length. To address this, we have incorporated relevant information about human models where it highlights key differences or contradictions with mouse findings.

We acknowledge the need for a comprehensive review focusing on human primitive endoderm development and are currently preparing such a review in our laboratory. To avoid overlap with the current review, our upcoming manuscript will provide a detailed analysis of human primitive endoderm models.

Page 6 line 124 of the the newly submitted manuscript:

The acquisition of pre-implantation-like XEN cells was achieved with the supplementation of medium with Activin A and WNT3a, which resulted in the generation of naïve, extra-embryonic endodermal progenitor (nEnd) cells³⁶. nEnd cell expression profile correlates with E3.5 and E4.5 PE, unlike XEN cells, which correlate with PE only at E4.5, and nEnd cells can contribute both to ParE and VE, indicating that chemically defined conditions can stabilize their lineage-derived functional characteristics. More recently, primitive endoderm stem cells (PrESCs), were efficiently derived from mouse blastocysts when the latter were cultured in the presence of GSK3 inhibiting conditions along with the FGF4 and PDGF-AA ligands, known to promote PE expansion. PrESCs capture the functionality of E4.5 PE better than XEN cells, since they efficiently incorporate in the PE layer once injected into blastocysts, as well as equally contribute to the PE-derived tissues, VE and ParE³⁷.

5. Lines 108-109: the term naïve is restricted to cells cultured in 2i-LIF. Cells cultured in LIF do not represent a naïve pluripotent state.

We respectfully disagree with the reviewer on this point. Silva et al. (2008) and Nichols et al. (2009) demonstrate that 2iLIF conditions represent the ground state of pluripotency. While serum/LIF conditions support a more heterogeneous culture, numerous publications also refer to serum/LIF-maintained mESCs as naïve, including Boroviak et al. (2014), Hackett (2014), Kalkan et al. (2017), and Ghimire et al. (2018). This is because serum/LIF mESCs still share key characteristics with the pre-implantation epiblast rather than the post-implantation epiblast.

6. Line 110: in the context of the in vivo counterpart of naïve ESCs Boroviak et al, Nature Cell Biology, 2014 needs to be mentioned.

We thank the reviewer, and we have included the reference mentioned above.

Page 5 line 109 of the newly submitted manuscript:

Transcriptional and functional analysis of mESCs, such as the ability to colonize embryonic, but not extra-embryonic lineages, ^{16,18,24,25} demonstrates their resemblance to pre-implantation EPI.

7. Line 112: there is no ICM at E4.5.

We thank the reviewer for the correction, and we have changed the text accordingly.

Page 5 line 111 of the newly submitted manuscript:

However, naive mESCs cultures in serum+LIF conditions are heterogeneous and mirror the E4.5 EPI cellular composition since they consist of dynamic subpopulations ²⁷⁻³².

8. Lines 122-123: it would be important to specify that XEN cells have an increased propensity to contribute to the parietal endoderm.

We have rephrased our text to state this information more clearly.

Page 5 line 120 of the newly submitted manuscript:

Precisely, XEN cells transcriptionally resemble the ParE more and predominately contribute to this layer ³³⁻³⁵. Yet, the culture of these cells in the presence of BMP4 drives them to a VE fate, indicating that the different stages of the extra-embryonic endoderm (ExEn) can be successfully captured *in vitro*.

9. Lines 146-147: Otx2 is also expressed in the visceral endoderm of the post-implantation mouse embryo (Acampora et al, Development, 2013).

We have included this information and reference in our text.

Page 7 line 156 of the newly submitted manuscript:

Interestingly, OTX2, typically a post-implantation EPI and VE marker in mice, is restricted in the human PE during the blastocyst stage ^{9,47,51}.

10. Lines 299-300: reference 78 needs to be more clearly explained, especially in the context of Silva et al, Cell, 2009. While GATA6 is expressed, other PE markers are not, and this needs to be mentioned.

We thank the reviewer for this remark, and we have explained reference 78 (now 74) better.

Page 13 line 309 of the newly submitted manuscript:

Conversely, *Nanog*^{-/-} embryos display pan-ICM expression of GATA6; *however, later PE markers are not expressed, indicating the non-cell-autonomous requirement from the EPI cells for PE maturation* (Table 1)⁷⁴.

11. Lines 308-318: this section is a bit confusing: first it is stated that FGF signaling is required for the initial GATA6 expression, and then that FGF is not needed for the initiation of PE fate but rather for maintenance. In this section, the concept that Nanog induces FGF4 expression and the phenotype of NANOG null ICMs could be presented.

We thank the reviewer for pointing out the lack of clarity. We have rephrased this part in the new manuscript. In addition, we have presented the concept that *Nanog* induces FGF4 expression. The phenotype of *Nanog* null embryos has already been presented.

Page 13 line 319 of the newly submitted manuscript:

Inhibition of RTK signaling via FGFR inhibition at the onset of *Gata6* expression (E2.5) resulted in no *Gata6*-expressing cells in the ICM⁷⁴. Similar results were observed in *Grb2*^{-/-} blastocysts, where *Gata6* expression was absent, and the ICM uniformly expressed *Nanog*⁸⁴. *However, the effects of chemical inhibition or genetic ablation were only investigated at late stages when lineages are specified or already segregated. Later studies established FGF signaling as the primary pathway that governs PE restriction and maintenance but not initiation.*

Page 14 line 348 of the newly submitted manuscript:

Initially, the process depends on the expression of *Fgf4* by a group of cells and, later, on the expression of *Fgfr2* by the PE-fated subpopulation. *It has been proposed that Fgf4 expression is activated by the stochastic coordinated expression of Nanog and other EPI markers in a group of cells at the 16-cell stage⁸⁹. At E3.5 (mid-stage 2), the co-expression of these factors enhances Fgf4 expression for proper EPI specification in the ICM, facilitating the PE specification of neighboring cells^{72-74,90,91} (Figure 3).*

12. Lines 327-329: here the authors explain that Fgfr1 expression in EPI progenitors maintains low Erk activity and Nanog maintenance, but in the next section they mention that Fgfr1 activation leads to Nanog downregulation. This is a bit confusing. What determines the two different outcomes?

We thank the reviewer for highlighting the lack of clarity in our explanation. We acknowledge that the confusion arose because we did not specify that the second outcome—activation of FGFR1 via FGF4 binding leading to *Nanog* downregulation—occurs at a later developmental stage (implanting blastocyst) rather than during EPI/PE specification.

Regarding EPI/PE specification, signaling through FGFR1 in EPI cells maintains low Erk activity through negative feedback involving ETVs, SPRYs, and/or DUSPs. It is plausible that these negative feedback loops are silenced during the implanting blastocyst stage, allowing FGF4 autocrine signaling through FGFR1 to diminish *Nanog* expression. This downregulation facilitates pluripotency exit in EPI cells and

promotes differentiation toward embryonic lineages (Molotkov et al., 2017; Lanner and Rossant, 2010). ESC differentiation assays support this theory (Molotkov et al., 2017); however, the precise mechanism has yet to be fully elucidated.

To address the reviewer's concern, we have revised the relevant section to clarify the timing of these events. Nevertheless, we prefer to omit speculative details in the current manuscript since the exact mechanism has not been established.

Page 22 line 527 of the newly submitted manuscript:

Later, during blastocyst implantation, FGF4 induces *Nanog* downregulation in EPI cells in a cell-autonomous manner through FGFR1, indicative of EPI lineage maturation, suggesting that FGF signaling is required for the EPI transition to a primed pluripotent state^{83,85,116}.

13. Line 338: what triggers Fgfr2 expression in the PE-biased population? If unknown it would be important to flag it.

We have pointed out this gap in the literature in the new manuscript.

Page 15 line 350 of the newly submitted manuscript:

It has been proposed that Fgf4 expression is activated by the stochastic coordinated expression of Nanog and other EPI markers in a group of cells at the 16-cell stage⁸⁹. At E3.5 (mid-stage 2), the co-expression of these factors enhances Fgf4 expression for proper EPI specification in the ICM, facilitating the PE specification of neighboring cells^{72-74,90,91} (Figure 3). Of note, the initial factors that trigger Fgfr2 expression remain unidentified.

14. Line 347: the human data appears in the middle of the mouse data. I would suggest making a specific section about human PE.

As mentioned previously, this review primarily focuses on the mouse model to provide a detailed description of the process of EPI versus PE specification, with relevant references to the human model where appropriate. While a dedicated section on human primitive endoderm development would be valuable, it falls outside the specific scope of this review. Nevertheless, we acknowledge the need for a comprehensive review on this topic in the context of the human model and we are currently preparing such a manuscript in our laboratory.

15. Line 460: this sentence is not clear. Does it mean that WNT9A localization is restricted to the blastocoel?

We rephrased the text for more clarity.

Page 20 line 476 of the newly submitted manuscript:

WNT9A, whose expression *is specifically detected in the cells surrounding the blastocoel cavity*, might be a potential regulator of gene expression stabilization from specification to segregation¹⁰⁶ (Figure 3).

December 19, 2024

RE: Life Science Alliance Manuscript #LSA-2024-03091-TR

Dr. Paraskevi Athanasouli
KU Leuven
Herestraat 49
Leuven, Vlaams-Brabant 3000
Belgium

Dear Dr. Athanasouli,

Thank you for submitting your revised manuscript entitled "Divergent destinies: Insights into the molecular mechanisms underlying EPI and PE fate determination". We would be happy to publish your paper in Life Science Alliance pending final revisions necessary to meet our formatting guidelines.

- please be sure that the authorship listing and order is correct
- please upload your figures as single files
- please add an Author Contributions section to your main manuscript text
- please add your figure and table legends to the main manuscript text after the References section

LSA now encourages authors to provide a 30-60 second video where the study is briefly explained. We will use these videos on social media to promote the published paper and the presenting author (for examples, see <https://docs.google.com/document/d/1-UWCfbE4pGcDdcgzcmiuJl2XMBJnxKYeqRvLLrLSo8s/edit?usp=sharing>). Corresponding or first-authors are welcome to submit the video. Please submit only one video per manuscript. The video can be emailed to contact@life-science-alliance.org

A. FINAL FILES:

B. MANUSCRIPT ORGANIZATION AND FORMATTING:

Sincerely,

December 23, 2024

RE: Life Science Alliance Manuscript #LSA-2024-03091-TRR

Dr. Paraskevi Athanasouli
KU Leuven
Herestraat 49
Leuven, Vlaams-Brabant 3000
Belgium

Dear Dr. Athanasouli,

Thank you for submitting your Review entitled "Divergent destinies: Insights into the molecular mechanisms underlying EPI and PE fate determination". It is a pleasure to let you know that your manuscript is now accepted for publication in Life Science Alliance. Congratulations on this interesting work.

DISTRIBUTION OF MATERIALS:

Again, congratulations on a very nice paper. I hope you found the review process to be constructive and are pleased with how the manuscript was handled editorially. We look forward to future exciting submissions from your lab.

Sincerely,
